# The Impacts of Invasive Crayfish and Other Non-Native Species on Native Freshwater Crayfish: A Review

**DOI:** 10.3390/biology13080610

**Published:** 2024-08-12

**Authors:** Sarah B. O’Hea Miller, Andrew R. Davis, Marian Y. L. Wong

**Affiliations:** School of Earth, Atmospheric and Life Sciences, University of Wollongong, Wollongong, NSW 2522, Australia

**Keywords:** introduced species, competition, predation, disease, reproductive impact

## Abstract

**Simple Summary:**

Freshwater crayfish play critical roles in their ecosystems; however, non-native species are increasingly threatening native crayfish populations worldwide. Here, we review the literature examining or documenting the impacts of invasive freshwater crayfish and other non-native species on native freshwater crayfish. Four key mechanisms through which invasive crayfish impact native crayfish were identified: competition, predation, introduction of disease, and reproductive impacts, while other non-native species primarily impact native crayfish through predation. We highlight a need for more field-based investigations examining competitive interactions between invasive and native crayfish and identify a deficit of monitoring efforts for many non-native species that are likely to substantially impact native crayfish populations. Overall, the findings underscore the profound and diverse threats posed by non-native species to native crayfish, necessitating further research to safeguard native crayfish populations.

**Abstract:**

Freshwater crayfish are vital species in ecosystems where they naturally occur, as they hold keystone and ecological engineering positions in these systems. Non-native species are common and widely spread throughout Earth’s freshwater ecosystems and can have severe impacts on native crayfish populations. There has yet to be a comprehensive global review of the impacts of non-native species on native crayfish. Two literature searches were conducted using Web of Science and Google Scholar to find articles to address four key aims: (1) summarise trends in the literature; (2) examine the mechanisms by which invasive crayfish impact native crayfish species; (3) examine the mechanisms by which other non-native species, such as fish, impact native crayfish species; and (4) identify gaps in knowledge and research priorities. This review highlights that a far greater amount of research has addressed the effects of invasive crayfish than other non-native species. The research on invasive crayfish focuses on four types of interactions with native crayfish: competition, predation, introduction of disease, and reproductive impacts. Studies addressing the impacts of other non-native species on crayfish indicate that predation and habitat destruction by these species are the key processes impacting native crayfish. It is evident that field-based research, particularly concerning competition between invasive and native crayfish, is limited. Therefore, further in situ research is needed to assess the validity of laboratory results in a natural setting. Additionally, in many cases, the impact of certain non-native species on native crayfish populations has gone unmonitored. For this reason, it is recommended that additional research focus on assessing the impact of these non-native species. To conclude, the impacts of invasive crayfish on native crayfish are profound and wide-ranging, often leading to population decline or extirpation. Further, other non-native species are also likely to have a highly deleterious impact on native crayfish populations; however, more research is required to understand the scope of this impact.

## 1. Introduction

Freshwater ecosystems support a disproportionate amount of biodiversity given the relatively small amount of area they occupy on Earth [1]. Alarmingly, freshwater biodiversity is declining at a considerably greater rate than that of terrestrial ecosystems due to habitat destruction, climate change, and invasive species [2]. Globally, invasive species have spread into freshwater systems to a greater extent than in terrestrial systems through various anthropogenic pathways [2] and are considered a leading threat to freshwater biodiversity and ecology [3,4].

The invasion process occurs through a series of stages, with many freshwater species possessing characteristics that enable their successful progression through these stages. The initial phase of the invasion process is the transportation and subsequent accidental or intentional introduction of a non-native species to a novel ecosystem [5,6,7]. Once a non-native species has been introduced, it must then become established as a self-sustaining population [8]. Following establishment, the non-native species spread from its initial point of introduction to expand its occupied range [6]. The life history characteristics of a species play key roles in the establishment and dispersal phases of the invasion process. Most successful invaders have some combination of high fecundity, asexual reproduction, a generalist diet, and broad environmental tolerance [3,6]. Additionally, since freshwater systems are susceptible to environmental extremes such as desiccation or freezing, many freshwater species have evolved characteristics that allow them to succeed under harsh environmental conditions [9,10]. The final stage of the invasion process involves the non-native species exerting negative ecological, economic, or human health-related impacts [6,11].

When a non-native species invades an ecosystem and establishes itself as an invasive species, it brings with it a wide range of ecological consequences [12]. The severity of these consequences is dependent on the nature of the invader; for example, the introduction of invasive ecosystem engineers can have far-reaching ecological impacts [13,14]. An ecosystem engineer is defined as a species that “directly or indirectly modulates the availability of resources to other species by causing physical state changes in biotic or abiotic materials” [15]. For example, the invasive freshwater golden mussel (*Limnoperna fortunei*) forms into beds, causing a physical change to the substratum. This, in turn, results in the increased abundance of certain species and the exclusion of others [16]. Therefore, invasive ecosystem engineers have the capacity to alter entire ecosystems [17]. Hence, a comprehensive understanding of their impacts is important. 

Freshwater crayfish are not only considered ecosystem engineers but also have keystone roles in the ecosystems to which they are native [18,19,20]. A keystone species is defined as an organism that has a disproportionate effect on the trophic structure or function of an ecosystem relative to its total biomass [21,22]. Crayfish perform keystone functions through their broad diet, which includes algae, macrophytes, macroinvertebrates, and vertebrates [18,23,24,25,26]. Therefore, they have a strong influence on the energy flow within an ecosystem. The ecological engineering roles of crayfish include sediment processing, burrow formation, and the breakdown of detritus [18,27,28,29]. As a result, they play an important role in nutrient cycling. For these reasons, freshwater crayfish hold a highly influential and crucial ecological position in the freshwater systems in which they naturally occur. 

The first accounts of crayfish introductions beyond their natural range date back to 1746 [30]. Since freshwater crayfish are both ecological engineers and keystone species, invasive crayfish are particularly problematic within the ecosystem. Their ability to occupy multiple trophic levels due to their omnivorous diets [31] means invasive crayfish can significantly disrupt the ecological functioning of a freshwater system [32,33,34]. In addition, invasive freshwater crayfish possess several key life history traits that contribute to their invasiveness. Most successful invasive freshwater crayfish exhibit early maturation, high fecundity, and rapid growth rates, as well as generalist and opportunistic feeding habits along with a tolerance of extreme environmental conditions [35,36,37,38]. Further, invasive crayfish commonly demonstrate greater aggression than their native counterparts, which also plays a key role in their invasion success [39,40,41,42,43]. 

These invasive life historyand behavioural characteristics, combined with their broad environmental tolerance and ecological engineering roles, have the capacity to result in severe direct and indirect effects on the survival and persistence of native crayfish species. Native species can be impacted by invasive crayfish directly through predation [44], competition for resources [45], and aggressive interactions [46]. Native crayfish can also be impacted indirectly via indirect competition [47], reproductive effects [48], and the introduction of disease [49]. In the past, these mechanisms have resulted in substantial and rapid declines in native crayfish populations [50,51] and even the complete exclusion or replacement of native crayfish by invasive crayfish species [52,53]. 

In addition to the ongoing threat posed by invasive crayfish, there are numerous other non-native species that have detrimental effects on native crayfish. Such species include introduced fishes such as brown trout [54] and terrestrial species such as American mink and foxes [55,56]. While these species directly impact native crayfish primarily through predation, certain species may also have indirect impacts through habitat destruction. For example, habitat destruction caused by the feeding habits of common carp (*Cyprinus carpio*) was reported as the most likely cause of the decline in native crayfish (*Cambarellus montezumae*) abundance in Mexico [57]. As such, these other non-native species may also contribute to the reduced abundance of native crayfish [54]. 

Owing to the widespread and ongoing problem of invasive crayfish globally, an increasing number of studies have investigated their impact on native crayfish (Figure 1). Previous reviews have synthesised the broad ecological impacts of invasive crayfish, with some mention of their impact on native crayfish [58,59,60]. Further, the threat of invasive crayfish to native crayfish in Europe has been reviewed [50,61]. However, such a review has yet to be undertaken on a global scale. Further, there has yet to be a review of the impacts of other non-native species on native crayfish. Therefore, this review seeks to address these knowledge gaps by (1) summarising trends in the literature; (2) examining the mechanisms by which invasive crayfish impact native crayfish species; (3) examining the mechanisms by which other introduced species impact native crayfish species; and (4) identifying further gaps in knowledge and research priorities.

## 2. Methods

### 2.1. Definitions

For this review, introduced species were defined as any species that has been introduced to an area in which it does not naturally occur. Invasive species were defined as those introduced to an area in which they do not naturally occur and where they exert negative ecological, and/or economic, and/or human health-related impacts [62]. A non-native species was defined as any species that is living outside of its natural range, i.e., this term encompasses introduced and invasive species. Some species may be considered native to a certain country but, because of human translocation, are defined as introduced or invasive species in areas within that country where they do not naturally occur. For example, in Australia, the common yabby (*Cherax destructor*) is native to the Murray–Darling Basin. However, their use in aquaculture and as bait by recreational fishers across the country has resulted in their proliferation as invasive species throughout waterways in Southeastern Australia and Western Australia. Here, native species were defined as species that occur in their natural range.

### 2.2. Search Procedures 

The literature for this review was searched using Web of Science and Google Scholar. To find articles either examining or documenting one or more impacts of an invasive freshwater crayfish on a native crayfish, the following search terms were used: (“invasive freshwater crayfish” or “introduced freshwater crayfish” or “alien crayfish” or “non-native freshwater crayfish”) AND (“impact” or “threat” or “compete” or “disease” or “predate” or “consume” or “reproductive interference” or “hybridization”) AND (“native freshwater crayfish” or “indigenous freshwater crayfish”). Only articles reporting results relevant to the review criteria (i.e., examining or documenting an impact of an invasive crayfish on a native crayfish) were included. The reference lists of all relevant articles found from the literature search were then examined to find any applicable articles that were missed. Each publication was then assigned to one of four categories based on the mechanism of impact the study assessed: (1)“Competition”—the study investigates competitive interactions between invasive and native crayfish.(2)“Predation”—the study investigates the impact of predation by invasive crayfish on native crayfish.(3)“Reproductive impacts”—the study investigates either reproductive interference or hybridisation by the invasive crayfish with native crayfish.(4)“Introduction of disease”—the study investigates the introduction of any disease by an invasive crayfish to a native crayfish population.

When a publication covered more than one area, it was assigned to multiple categories. 

To find articles examining or documenting the impacts of other introduced species on native crayfish, the following search terms were used: (“invasive species” or “introduced species” or “alien species” or “non-native species”) AND (“impact” or “compete” or “predate” or “consume” or “habitat destruction”) AND (“native freshwater crayfish” or “indigenous freshwater crayfish” or “crayfish”). Only articles reporting results relevant to the review criteria were included. The reference lists of all relevant articles found from the literature search were then examined to find any articles that were missed. 

## 3. Results and Discussion

### 3.1. Trends in the Literature

A total of 83 journal articles were identified as examining or documenting an impact of invasive crayfish species on native crayfish species. Of these studies, 52% were focused on invasive crayfish from the Cambaridae family, 33% on Astacidae, and 15% on Parastacidae (Figure 2). It is apparent that certain invasive species have been more extensively studied, the top four being *Pacifastacus leniusculus*, *Faxonius rusticus* (formerly *Orconectes rusticus*), *Procambarus clarkii*, and *Cherax destructor* (Figure 2). This is a likely consequence of how widespread these invasive species are relative to other invasive crayfish. Articles that examined interspecific competition between invasive and native crayfish were most common, followed by the introduction of disease, reproductive impacts, and predation, respectively (Figure 3). Investigations of the impact of invasive crayfish on native crayfish in European countries accounted for 45% of studies, followed by North America (31%) (Figure 4). The remaining 24% comprised studies from other continents (Figure 4). This distribution of the literature across the continents is likely related to both the number and abundance of native crayfish species, the presence of invasive crayfish on each continent, and the number of mechanisms through which the invasive crayfish impact natives. For example, in Africa, only seven native crayfish species occur, all of which are isolated to Madagascar, and only one invasive crayfish, *Procambarus virginalis*, has been introduced to Madagascar [63]. Therefore, the literature examining the impacts of invasive crayfish on native crayfish in Africa is limited (Figure 4). Of the European studies, the impact of the introduction of disease by invasive crayfish was the focus of 61% of articles, whereas the North American studies focused predominantly on the effects of competition (54%) and reproductive impacts (42%) (Figure 4). 

Overall, 33 journal articles were identified as examining or documenting the impact of other non-native species on native crayfish species. Of these, 58% were focused on terrestrial species and 42% on aquatic or amphibious species (Figure 5). The two primary threat processes identified by these articles were predation by non-native species on native crayfish and the destruction of important habitats. Additionally, one article examined the spread of the crayfish plague by the invasive Chinese mitten crab (*Eriocheir sinensis*) and another in American mink feces. Of the articles conducted on the terrestrial species, American mink (*Neovison vison*) was the focus of 53%, with the remaining documenting the impacts of Northern raccoon (*Procyon lotor*) (20%), red fox (*Vulpes vulpes*) (20%), and wild pig (*Sus scrofa*) (7%) (Figure 5). The literature relating to non-native aquatic or amphibious species was largely focused on five species: brown trout (*Salmo trutta*), rainbow trout (*Oncorhynchus mykiss*), common carp (*Cyprinus carpio*), redfin perch (*Perca fluviatilis*), and African jewelfish (*Hemichromis bimaculatus*) (Figure 5). 

Overall, it is apparent that studies examining or documenting the impact of invasive crayfish on native crayfish are far more prevalent in the literature, with a total of 83 studies found on the topic, over two times the number of articles investigating the impacts of other non-native species. This pattern could be attributed to the wide range of threats posed by invasive crayfish, i.e., competition, predation, reproductive impacts, and the introduction of disease, compared to the relatively fewer perceived threats posed by other invasive species, i.e., predation and habitat destruction. It may also be the case that examining the impact of taxa such as fish and mammals may present more challenges than invasive crayfish since the latter are typically easier to trap and work with. 

### 3.2. The Impacts of Invasive Crayfish on Native Crayfish

#### 3.2.1. Competition

Competitive interactions between invasive and native crayfish can result in injury to natives [52] and reduced access to limited resources [64]. This, in turn, can lead to reduced abundance [65,66] and possibly extirpation of the native species in the most extreme cases. Indirect evidence for competition comes from stable isotope and gut contents analyses, which have revealed overlaps in the diet composition or trophic position of native and invasive crayfish when living in sympatry [47,67,68,69]. Further, studies using radio telemetry to determine habitat use in sympatric invasive and native crayfish have reported overlap in the refuge preferences of each species [70,71], suggesting that competition for shelter between invasive and native crayfish is likely. 

When resources overlap and competition is likely, laboratory studies have commonly reported that invasive crayfish are competitively dominant over native crayfish. For example, the invasive *Faxonius rusticus* has been found to readily dominate the native *F. virilis* in staged interspecific contests [45,72,73]. This pattern of invasive dominance has also been reported in *F. limosus* [74], *Pacifastacus leniusculus* [75,76,77], *Procambarus clarkii* [78], *Cherax destructor* [39,43,79] and *Cherax quadricarinatus* [80]. Invasive dominance has also been observed in situ. For instance, in Swedish and Finnish lakes where the invasive *Pacifastacus leniusculus* has spread, the number of native *Astacus astacus* possessing chelae injuries is significantly greater than the number of *P. leniusculus* showing the same type of injury. This asymmetry suggests that the invasive *P. leniusculus* has a competitive advantage during aggressive interactions [52,65]. In another study, the presence of invasive *Faxonius rusticus* was related to a shift in the habitat associations of native crayfish, which occupied less-preferred habitat types (i.e., sand and macrophytes) in their presence compared to the preferred cobble habitat occupied when isolated from *F. rusticus* [64]. This provides further support for the competitive dominance of the invasive species, resulting in the exclusion of the native species from specific habitat types.

The exclusion of native crayfish from certain habitats and refuges by invasive crayfish can lead to an increased susceptibility of natives to predation. For example, Garvey et al. [46] reported that the invasive *Faxonius rusticus* excludes the native *F. virilis* from shelters, resulting in increased predation by largemouth bass. The same pattern was observed between invasive *Pacifastacus leniusculus* and native *Astacus astacus*, which was consumed by European perch [81]. Differential predation by fish can also have non-consumptive effects on native species. For example, *F. virilis* experienced a significantly diminished growth rate and increased non-consumptive mortality in the presence of largemouth bass, whereas invasive *F. rusticus* and *F. propinquus* experienced a much lesser effect [82]. 

There are several factors that cause asymmetries in the fighting abilities of invasive and native crayfish and thus account for the competitive dominance of invasive species. Relative aggressiveness plays a key role in the outcome of agonistic interactions, and invasive crayfish often demonstrate greater intrinsic aggression than their native counterparts [45,46,76]. In addition, differential fighting dynamics can result in the competitive advantage of the invasive species. For instance, a comparison of intraspecific pairs of invasive *Pacifastacus leniusculus* and native *Astacus leptodactylus* reported that fights between *P. leniusculus* were significantly longer than the fights between natives. In interspecific interactions, the ability of *P. leniusculus* to fight for a longer duration resulted in their dominance over *A. leptodactylus* [83]. Finally, size differences between native and invasive crayfish often translate to dominance over the naturally smaller species. For example, invasive *P. clarkii* has a larger body and chelae size than the native *Austropotamobius italicus*, resulting in the greater fighting ability of the invader [84]. 

Although most of the literature reports invasive crayfish as competitively dominant over natives, there are some exceptions to this finding. In staged interactions between the invasive *Pacifastacus leniusculus* and the native *Austropotamobius torrentium*, no pattern was found in the competitive dominance of either species [85]. Additionally, in staged agonistic interactions over shelter between the invasive *Faxonius limosus*, *Procambarus acutus*, and the native *Astacus astacus*, *A. astacus* was the dominant species [86]. Further, in staged agonistic interactions over a food resource between size-matched invasive *Cherax destructor* and native crayfish, the native *Euastacus dharawalus* demonstrated more aggressive behaviours, was less submissive than *C. destructor*, and won significantly more contests than *C. destructor* [87]. In contrast, in situ observations of competitive interactions between *E. dharawalus* and *C. destructor* established that when *E. dharawalus* was the larger competitor, the species was significantly more likely to win. However, when size-matched, neither species exhibited a significant pattern of dominance [88]. Other field-based studies have found a lack of competitive dominance for invasive crayfish over natives. In situ competition experiments between invasive *Faxonius neglectus* and native *F. eupunctus* revealed that the growth and survival of the natives were not impacted by the presence of the invasive species [89]. Moreover, field surveys of invasive *F. hylas* and native *F. quadruncus* demonstrated no shift in the habitat use of the native species when in sympatry with the invasive [90], indicating neither species is inherently competitively dominant over the other. 

In addition to the various traits that influence competitive dominance, it is increasingly being recognised that various abiotic factors can modulate competition between invasive and native crayfish. For instance, contests performed under current-day temperatures (22 °C) between invasive *Cherax destructor* and native *Euastacus spinifer* established that *E. spinifer* was more likely to win. However, under near-future predicted temperatures (26 °C), *C. destructor* obtained a competitive advantage and was more likely to win contests [91]. This advantage is likely attributed to the increased thermal tolerance of *C. destructor*, as this species is adapted to warmer conditions in contrast to *E. spinifer*, which is adapted to cooler conditions (under 24 °C). Further, a recent study by Marn et al. [92] used Dynamic Energy Budget models to assess the response of European native and invasive crayfish to conditions likely to arise from climate change. In the invasive *Procambarus virginalis*, they found substantial increases in maturation rate and reproductive output with increasing water temperature, increasing the invasion potential of this species. We suggest that future elevated water temperatures resulting from climate change will likely result in some species of invasive crayfish obtaining a competitive advantage over natives. Additionally, flow discharge is reported to influence competition for food and shelter between the invasive *Faxonius rusticus* and native *Faxonius virilis*. *Faxonius rusticus* was significantly more likely to win fights over food and shelter resources in the high-discharge treatment, whereas *F. virilis* was significantly more likely to win fights over food but not shelter in the low-discharge treatment [93]. Therefore, flow is likely to influence the outcomes of competitive interactions between these species in situ. 

#### 3.2.2. Predation

Predation by invasive crayfish has been cited as an important mechanism contributing to the exclusion of native crayfish [52,94]. However, there has been very limited research confirming the impact of predation on native crayfish. To our knowledge, there have only been two laboratory-based studies examining this effect. Holdich et al. [95] studied intraspecific and interspecific interactions between two invasive crayfish, *Pacifastacus leniusculus* and *Astacus leptodactylus*, and the native *Austropotamobius pallipes*. While there was little mutual predation in interspecific groups of adult *A. leptodactylus* and *A. pallipes*, in interspecific groups of adult *P. leniusculus* and *A. pallipes*, there was significant predation pressure by *P. leniusculus.* Similarly, Nakata and Goshima [44] found that heterospecific predation by *P. leniusculus* on native *Cambaroides japonicus* occurred at a far greater rate than any heterospecific predation by *C. japonicus.* Predation by invasive crayfish on native crayfish has yet to be confirmed in situ. This is an important area of research, given multiple reports of invasive crayfish exhibiting strong intraspecific cannibalism in natural settings (*P. clarkii*: [96]; *P. leniusculus*: [97,98]); therefore, in situ predation by invasive crayfish on both juvenile and adult native crayfish is likely. 

#### 3.2.3. Reproductive Effects 

Reproductive and genetic interference with native species are significant risks associated with the introduction of invasive crayfish. Reproductive interference involves a native species either accidentally or preferentially mating with invasive counterparts. For threatened native species, the consequences of this can be especially severe, as the gene flow between already fragmented populations may be disturbed, resulting in genetic bottlenecks [58]. If an invasive and native species can successfully reproduce, this results in the production of hybrids, which could present several risks to native populations. For instance, hybrids may be sterile, resulting in a waste of gametes and a genetic ‘dead end’ for natives. Further, the production of hybrids may result in another invasive genotype, which may compete with natives for resources, causing further displacement. Finally, hybridisation introgression may occur, causing widespread ‘genetic pollution’ within the native population [99]. 

There is some evidence to suggest that preferential mating with invasive species may occur and play a key role in the displacement of certain native crayfish species. Mate selection experiments performed between invasive *Faxonius rusticus* and native *Faxonius sanborni* revealed both male *F. rusticus* and male *F. sanborni* preferentially mated with *F. rusticus* females [100]. The same outcome was found in mate-selection tests between *F. rusticus* and *F. propinquus*, with males of both species choosing *F. rusticus* females over *F. propinquus* females [101]. It may be the case that some native species are unable to differentiate between the chemical signals produced by their mates and those of congeners [101]. Further, because crayfish tend to choose similar-sized or larger-bodied mates [102], and *F. rusticus* is naturally larger in size, these native species preferentially select the largest available mates. 

In some cases, reproductive interference resulting from preferential mating can lead to hybridisation between native and invasive species. Hybridisation between *Faxonius* species is well documented in the literature. Laboratory-induced hybridisation between invasive *F. rusticus* and native *F. propinquus* has been achieved [103], and at sympatric sites, morphological and genetic evidence indicates that hybridisation between these species is common [48,104,105]. In an extensive genetic study, Perry et al. [48] reported that 95% of hybrids collected from a northern Wisconsin lake (USA) were the product of interbreeding between *F. propinquus* males and *F. rusticus* females, in accordance with the mate-selection test conducted by Tierney and Dunham [101]. It was also apparent that a significantly greater proportion of the population sampled was a product of F_1_ hybrids backcrossing with *F. rusticus* than F_1_ hybrids backcrossing with *F. propinquus.* This indicates F_1_ hybrids may preferentially mate with *F. rusticus*, thus enhancing the effect of hybridisation introgression on the *F. propinquus* population. In addition to *F. propinquus*, there has been morphological evidence collected in situ that suggests *F. rusticus* is able to hybridise with *F. limosus* [106] but not with *F. virilis* [48]. Recently, Rozansky et al. [107] reported that the invasive populations of *F. virilis* have hybridised with the native *Faxonius punctimanus*. 

Outside of the *Faxonius* genus, there is little mention of hybridisation in other native–invasive crayfish systems. The only other instance of invasive–native hybridisation was reported to occur between invasive *Cherax cainii* and native *Cherax tenuimanus*, historically considered a single species [108]. Austin and Ryan [109] reported evidence of morphological and genetically intermediate hybrids between the two species, which was supported by further genetic research [110,111]. However, an investigation into the extent of hybridisation between *C. cainii* and *C. tenuimanus* revealed the levels of introgression were significantly lower than expected under random mating; therefore, there are likely reproductive barriers in place between these two species [112]. In general, there may be few examples of hybridisation because the native–invasive species examined in other studies are from different genera and, therefore, unlikely to produce fertile offspring even if mating did occur. 

#### 3.2.4. Introduction of Disease 

The proliferation of North American crayfish as an invasive species has resulted in the spread of the oomycete, *Aphanomyces astaci*, the cause of the ‘crayfish plague’ [113]. As *A. astaci* co-evolved with North American crayfish, it typically acts as a benign parasite in this species. However, in crayfish not originating from North America, infection by the crayfish plague can be lethal. This differential susceptibility to the disease may be a key mechanism underlying the displacement of native European crayfish by invasive species. *Pacifastacus leniusculus*, *Procambarus clarkii*, and *Faxonius limosus* are the primary invasive species acting as vectors of *A. astaci*; however, it does not need a host to spread. Damp traps, nets, and cages can also provide a surface the spores can attach to [114,115]; thus, the crayfish plague can also be transmitted via the movements of fishers. Unfortunately, this disease has now spread widely throughout Europe [113] and has also been detected in native crayfish populations in South America [116,117] and Asia [118]. 

The spread of *A. astaci* in Europe has resulted in mass mortalities in multiple native crayfish populations. For example, in Turkey, the harvest of *Astacus leptodactylus* in 1984 reached 5000 metric tonnes; this decreased to 320 metric tonnes in 1991 after the plague was introduced into Turkish waterways [119]. Significant declines in native populations attributed to the crayfish plague have also been documented in Spain [49,120], Sweden [94], Germany [121], Estonia [122], Italy [123], the Czech Republic [124], and Ireland [114]. However, recovery among native crayfish populations is possible, with surviving populations of indigenous crayfish being found at sites of previous mass mortalities in the Czech Republic [124]. Despite this, the crayfish plague still represents one of the most significant threats to indigenous European crayfish. 

Outside of Europe, the risk posed by *A. astaci* to indigenous crayfish is also high. In Japan, two cases of mass mortalities in populations of the native *Cambaroides japonicus* were found to be a result of a crayfish plague outbreak originating from introduced *Procambarus clarkii* populations [118]. In South America, *A. astaci* was recently detected for the first time in indigenous South American crayfish populations and introduced to populations of *P. clarkii*. However, in this case, no significant native population declines have been documented [116]. Further, infection of introduced captive and wild populations of *P. clarkii* in Indonesia has also been confirmed; however, there is no evidence to suggest it has spread to native crayfish yet [125]. Laboratory research reported that Australian crayfish species are highly susceptible to *A. astaci* [126]; therefore, its introduction to Australia would have severe consequences for native Australian crayfish. 

Other diseases transmitted by invasive crayfish have been overshadowed by the crayfish plague due to its severity. For this reason, there is significantly less research on other pathogens transmitted by invasive crayfish. One of these is the protistan crayfish parasite *Psorospermium haeckeli*. Although little is known about the pathogenic impact of this parasite, it has been detected across all continents with indigenous crayfish and has been linked to mortalities in freshwater crayfish [127]. The invasive *Pacifastacus leniusculus* has been identified as a vector of *P. haeckeli* and is the likely cause of the spread of the parasite in Europe [128]. In Europe, mass mortalities of native crayfish have been attributed to the spread of the microsporidian parasite *Thelohania contejeani* [129], the causative agent of porcelain disease. This disease has been detected in invasive *P. leniusculus* [130], and it has been suggested that this species may act as a reservoir for *T. contejeani* [129]. Recent research in North America has reported a novel microsporidium parasite, *Cambaraspora faxoni*, within the native range and invaded range of *Faxonius rusticus* and the native range of *F. virilis*, where this species is in sympatry with invasive *F. rusticus* [131]. Stratton et al. [131] hypothesise that *C. faxoni* may have been introduced to the *F. virilis* populations with the invasion of *F. rusticus*. Alternatively, it has been suggested that the parasite may be native to both *F. rusticus* and *F. virilis* or that *C. faxoni* infects an alternative host with an overlapping range with *F. rusticus* and *F. virilis*. Since *C. faxoni* infection causes muscular degeneration in the host, infected *Faxonius* crayfish would likely be disadvantaged in competition with other crayfish.

Invasive crayfish may also bring with them their symbionts, which, when attached to native crayfish, could have a detrimental impact on their health. In Spain, a non-native ectosymbiont worm, *Xironogiton victoriensis*, was found on a native population of the European crayfish, *Austropotamobius pallipes*. This symbiont was likely introduced via translocations of the invasive *P. leniusculus* throughout the country [132]. The effect of *X. victoriensis* on native crayfish is unknown; hence, more research into this area is required. 

### 3.3. The Impact of Other Non-Native Species on Native Crayfish

#### 3.3.1. Non-Native Aquatic and Amphibious Species

Introductions of non-native aquatic species have occurred on all continents where endemic crayfish are found. Despite this, the impacts of non-native aquatic species on crayfish populations are understudied, and in the case of most species, it has not yet been established if they have an impact on native crayfish. Most research in this area has focused on three species: common carp (*Cyprinus carpio*), brown trout (*Salmo trutta*), and redfin perch (*Perca fluviatilis*). For this reason, most of the evidence provided in this section will draw from research on these species. 

Common carp is one of the most widely spread invasive freshwater fish species globally, yet there has been limited research on their influence on native crayfish populations. Crayfish have been reported to form some part of the carp diet when found in sympatry [133,134]. However, only one study to date has investigated the ecological interactions between invasive carp and native crayfish. Hinojosa-Garro and Zambrano [57] examined the underlying mechanism causing a decreased abundance of the native crayfish (*Cambarellus montezumae*) in freshwater systems in Mexico when carp were present. The gut content analysis did not indicate that direct predation by carp was a key factor in reducing crayfish abundance, suggesting that habitat depletion by carp is the more likely cause of this decline. Carp density has been reported to negatively correlate with submerged macrophyte cover [135], which is either caused by direct destruction of the macrophytes [136] or increased turbidity stemming from the bottom-feeding habits of carp [137], which in turn reduces light penetration and reduces macrophyte cover. Low macrophyte cover may mean reduced food resources for crayfish or reduced cover, causing an increased susceptibility to predation by other organisms. Hinojosa-Garro and Zambrano [57] also reported altered crayfish behaviour, with crayfish moving significantly faster in the presence of carp. This could cause crayfish to allocate more energy to seeking refuge rather than spending energy on growth, reproduction, or searching for food. 

Brown trout were introduced to New Zealand in the 1860s for the purpose of establishing a recreational fishery [138]. Currently, brown trout have proliferated throughout New Zealand’s freshwater systems and are found in high abundance in the rivers or lakes where they have been established. Field research has found the presence of the native crayfish *Paranephrops planifrons* is negatively correlated with the presence of brown trout [54,139]. Further, Olsson et al. [140] reported that crayfish abundance was significantly lower in streams with brown trout than in streams where trout were not found. It is likely these findings are associated with trout predation on the crayfish, as a gut content analysis conducted in the same study reported that a large trout had consumed a 5 cm (total length) crayfish. Additionally, it is likely that *Paranephrops planifrons* are more susceptible to predation by brown trout than other native predators because of their inability to detect the chemical cues produced by trout [141]. 

In Australia, brown trout, brook trout (*Salvelinus fontinalis*), and rainbow trout (*Oncorhynchus mykiss*) are stocked in many freshwater systems across the country. In Western Australia, the stomach contents of both brown trout and rainbow trout were found to contain native freshwater crayfish, *Cherax albidus* and *C. quadricarinatus* [142]. Further, one study has revealed rainbow trout heavily predate native *C. cainii* populations in Western Australia. Analysis of the stomach contents of the rainbow trout found *C. cainii* constituted 61% by volume of the trout stomachs examined. Further, stable isotope analysis found *C. cainii* formed most of the assimilated diet of large rainbow trout [143]. These findings indicate *C. cainii* populations may be seriously impacted by rainbow trout predation. In addition, an introduced population of *C. destructor* in a New South Wales Dam has been reported to be the major dietary source of stocked brown and rainbow trout [144]. Therefore, it is likely these species of trout are consuming native crayfish populations in areas where they have been stocked. 

In Australia, redfin perch has become widespread across cooler regions since its introduction in the 1860s. The carnivorous nature of this species means it has the capacity to consume large numbers of native freshwater species. For example, the native crayfish *Cherax tenuimanus* was found to make up between 36% and 90% of the diet in adult perch in Western Australia [145]. Further, another study reported that *C. tenuimanus* constituted between 55% and 88% of the volume consumed by adult perch in spring, summer, and autumn [146]. Morgan et al. [145] also reported that numerous stomach samples from perch contained appendages from large *C. tenuimanus*, indicating the ability of the perch to injure adult *C. tenuimanus.* Based on the findings of these studies, it is likely redfin perch is having a severe impact on *C. tenuimanus* populations; however, further research is required to understand the scope of this impact. 

In Florida wetlands, the introduced African jewelfish (*Hemichromis bimaculatus*) and, more recently, the Asian swamp eel (*Monopterus albus*) have become problematic invasive species. Surveys across 45 Florida wetlands recorded a greater catch-per-unit effort for numerous aquatic species, including the native freshwater crayfish *Procambarus alleni*, at sites where the African jewelfish was present than at sites where the species was absent [147]. Recently, Pintar et al. [148] reported that African jewelfish presence during a ‘boom’ period was associated with significantly lower densities of native *P. alleni* and *Procambarus fallax*. Additionally, the presence of the Asian swamp eel was significantly negatively associated with the density of *P. alleni* and *P. fallax*. These outcomes suggest that both African jewelfish and Asian swamp eel have significant predatory impacts on native freshwater crayfish in this area. 

The freshwater fish, *Channa maculata*, or Asian snakehead, was introduced to Madagascar in 1978. This predatory species represents a significant threat to native Madagascan fish, with many species experiencing steep population declines since its introduction [149]. It is likely that Asian snakeheads also pose a considerable threat to the seven endemic species of freshwater crayfish in Madagascar, particularly *Astacoides betsiloensis*, as it has been established that they consume juveniles of this species [150]. 

The extent of research surrounding the impacts of non-native aquatic species other than introduced freshwater fish on native crayfish is very limited. One study examined the capacity of the invasive Chinese mitten crab (*Eriocheir sinensis*) to transmit the crayfish plague pathogen, *Aphanomyces astaci*, to the native European crayfish, *Astacus astacus* [151]. It was established that transmission of *A. astaci* between these species is possible. This outcome is highly concerning given that the Chinese mitten crab has spread widely as an invader across Europe and North America and has the capacity to spread even further in these regions [152]. 

Finally, with the proliferation of cane toads (*Rhinella marina*) throughout Australia, this species is considered a threat to Australian freshwater crayfish [153]. However, Crossland [154] reported that the consumption of early, mid, and late developmental stage cane toad larvae had no apparent adverse effect on the Australian crayfish, *Cherax quadricarinatus*. However, as is the case with most of the non-native species mentioned above, this is an area of very limited research. Therefore, further investigation of the resistance of other crayfish species to cane toad toxin is required to determine the scope of this species’ impact on freshwater crayfish. 

#### 3.3.2. Non-Native Terrestrial Species

The American mink (*Neovison vison*) was introduced into Europe for use in commercial fur farms in the 1920s. Escapees established a feral population that has spread throughout at least 28 European countries [155]. Alongside other ecological impacts, including competition with the native Eurasian otter (*Lutra lutra*), American mink are known to consume native crayfish. In Sweden and Ireland, scat analyses have indicated native crayfish make up a significant proportion of the minks’ diet, especially in warmer months when crayfish are more active [156,157]. Crayfish remains have also been found in the scats of American mink populations in Poland, although this contribution was minimal [158]. However, stable isotope analysis revealed crayfish constituted ~40% of the mink diet in Spain [159], and in the Czech Republic, the native stone crayfish (*Austropotamobius torrentium*) was represented in 82% of all the mink scats examined [55]. It was also reported that mink selectively prey on sexually mature crayfish, affecting the reproductive potential of the population [55]. Yanuta et al. [160] reported that American mink scat analysis serves as an effective bioindicator of the presence and density of the invasive crayfish, *Faxonius limosus*. Given that American mink has continued to expand its range in Europe [161], this species continues to represent a substantial predatory threat to native European crayfish. 

Another non-native terrestrial species attracting substantial attention in Europe and Japan in recent years is the Northern raccoon (*Procyon lotor*). This species was introduced to Europe and Japan through their use in the fur and pet trade and has recently undergone rapid range expansion in both areas [162,163]. In Italy, Tricarico et al. [164] reported that populations of the native crayfish, *Austropotamobius pallipes*, were significantly reduced or absent in areas where raccoons were present. Further, Boncompagni et al. [165] reported that crustaceans constituted 60% of the total diet of raccoons in Italy and found numerous *A. pallipes* carcasses that showed signs of raccoon predation. However, in Japan, it is evident that raccoons consume crayfish to a lesser extent, with no crayfish found in the species gut contents, and stable isotope analysis indicated that crustaceans only make up ~5% of the raccoon diet [166]. Nonetheless, it is evident that the Northern raccoon represents a considerable predatory threat to native crayfish in countries where it has spread as an invader. 

In Australia, the introduced red fox (*Vulpes vulpes*) has become widespread and is considered a serious threat to native wildlife. Fox scat analyses conducted in Victoria and New South Wales revealed native crayfish were common in the scats in warmer months, likely due to increased crayfish activity and the tendency of the crustaceans to forage in terrestrial habitats [56,167]. Further, stomach contents analysis of foxes in Queensland reported crayfish contributed a small part of their diet [168]. In addition to the red fox, invasive wild pigs (*Sus scrofa*) in Australia may also pose a significant threat to native crayfish through habitat destruction. McCormack [169] reported that essential habitat for juvenile crayfish can be completely excavated via the feeding activity of wild pigs. However, it is unknown whether this activity negatively impacts the abundance of native crayfish. 

### 3.4. Knowledge Gaps and Future Research

A summary of the current knowledge surrounding the impact of invasive crayfish and other non-native species on native crayfish, as well as the knowledge gaps identified as a result of this literature review, is provided in Figure 6 for invasive crayfish and Figure 7 for other non-native species. It is evident that the most extensively studied interaction between invasive and native crayfish is competition; however, these interactions have been studied to a greater extent in laboratory-based experiments rather than in more natural field settings. Although laboratory research is important for isolating the effects of competition from other factors, the results may not always translate to competitive outcomes and dynamics in the field. Of the observations of competition between crayfish observed in field settings, these interactions are shorter, less intense, and modulated by various extrinsic factors [170,171]. An example of laboratory versus field disparity is apparent in the outcomes of a laboratory study by Lopez et al. [87] and a field-based study by O’Hea Miller et al. [88]. Both studies examined size-matched competitive interactions between the native *Euastacus dharawalus* and the invasive *Cherax destructor*. While Lopez et al. [87] report the native *E. dharawalus* as the dominant competitor in staged interactions against *C. destructor*, O’Hea Miller et al. [88] reported neither species to be significantly more dominant than the other when observed in situ using baited remote underwater video. Therefore, in situ observational studies are important to provide validity to laboratory study findings and to determine how they differ in a natural setting. 

In contrast to invasive–native crayfish competition, there has been a paucity of research on the impact of predation by invasive crayfish on native crayfish. The lack of research is concerning due to the cannibalistic nature of certain species of invasive crayfish [96,97,98]. Specifically, a focus on the rates of juvenile or young-of-the-year native crayfish consumption by invasive crayfish is needed since this is a likely hidden threat that appears to have received no research to date. Hence, it is imperative that more studies focus on utilising stable isotopes, gut content analysis, or direct observations to establish the prevalence and severity of predation by invasive species on native crayfish. There is also a paucity of research on other potential sources of disease brought by invasive crayfish other than the crayfish plague, particularly in Europe. Disease has the potential to silently wipe out entire populations of native crayfish; therefore, attention must be paid to pathogens detected in invasive crayfish populations, and it is critical that their impact on native crayfish is assessed. 

This review has also highlighted the relative paucity of research on the impacts of other non-native species on native crayfish. Most studies to date document their consumption of native crayfish, but few have examined the impact of predation at the population level. Importantly, the impact of many widespread and commonly introduced non-native species, including trout (rainbow, *Oncorhynchus mykiss*, brown *Salmo trutta*, and brook *Salvelinus fontinalis*), has yet to be determined. For example, in the past 20 years, 63,700,000 trout have been stocked into New South Wales (Australia) waterways to maintain a recreational fishery [172], yet despite the predatory nature of these species [143,144] and the presence of over 30 endemic species of freshwater crayfish in the New South Wales region [169], many of which have threatened status, their potential impact on native crayfish populations remains unknown. Furthermore, this review found no studies examining the potential impact of non-native plant species on freshwater crayfish. This may be an area of considerable future study, considering the vast number of invasive aquatic plants and the degree to which some species can alter a freshwater ecosystem. For example, willows (*Salix* spp.) can dramatically alter flow regimes and erosion [173] as well as significantly reduce instream macroinvertebrate abundance [174]. 

The mechanisms by which invasive crayfish and other non-native species impact native crayfish populations may be further exacerbated by the effects of climate change. Some invasive species may thrive under altered conditions and, therefore, increase their impact on native species [91,175]. Further, native species may experience heightened pressure because of unfavourable conditions [176]. More research in this area is essential for predicting how interactions between invasive species and climate change will affect native freshwater crayfish.

## 4. Conclusions

The evidence synthesised in this review underscores that the spread of invasive freshwater crayfish is of great concern due to the number of pathways by which they may impact native freshwater crayfish. Due to the diverse and significant effects that invasive crayfish can have on native crayfish, we recommend more field-based research to better explore and monitor the impacts of invasive crayfish on native populations. Additionally, it is important that careful consideration is given to understanding the potential effects on native crayfish prior to any further non-native crayfish translocations. The literature reviewed here also suggests that non-native fish and terrestrial species have significant potential to have severe impacts on crayfish populations. Despite this, the introduction of fish into freshwater ecosystems for use in recreational fishing continues in many areas. It is important that these introductions be assessed for their influence on natives. Further, prior to new introductions, more research is required to understand the scope of the potential effects of non-native fish species. There is also room for considerably more research into the interactions between future conditions resulting from climate change and invasive species and how this interaction might affect native freshwater crayfish. 

## Figures and Tables

**Figure 1 biology-13-00610-f001:**
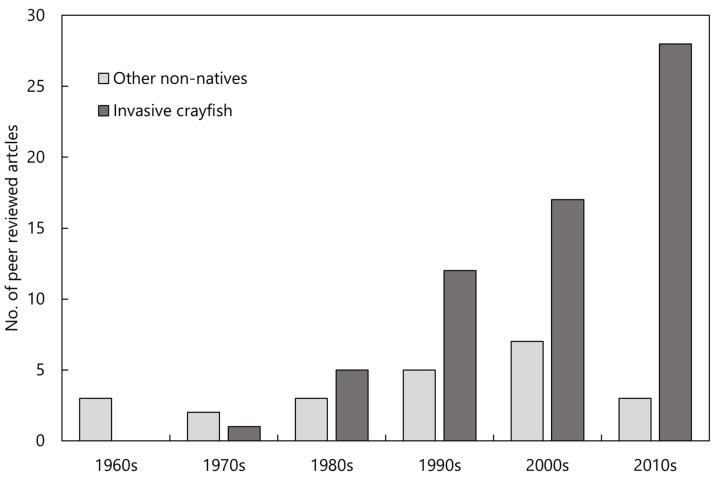
The number of peer-reviewed articles published on the impact of invasive crayfish and other non-native species of native freshwater crayfish within each decade since the 1960s. Articles tallied for this figure were found using the search method described in the Section 2.

**Figure 2 biology-13-00610-f002:**
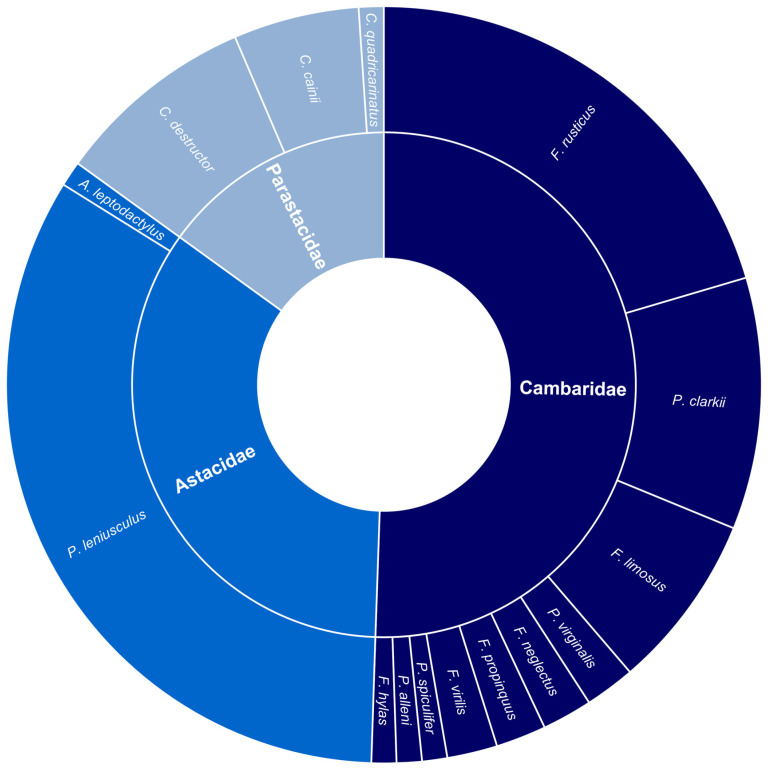
Summary of the literature investigating the impacts of invasive freshwater crayfish on native freshwater crayfish (*n* = 83). The inner ring quantifies the proportion of each invasive freshwater crayfish family appearing in the literature. The outer ring indicates the proportion of the literature focused on each invasive crayfish species. In Parastacidae, C. = *Cherax*; in *Cambaridae*, F. = *Faxonius* and P. = *Procambarus*; in *Astacidae*, P. = *Pacifastacus* and A. = *Astacus*.

**Figure 3 biology-13-00610-f003:**
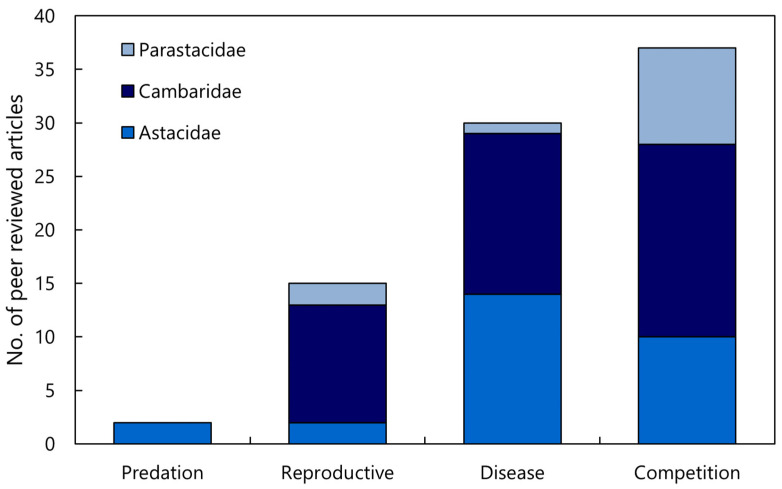
Analysis of the literature investigating the impact of invasive crayfish on native crayfish across the four mechanisms of impact and families (*n* = 83). If a research article included multiple families, they scored one for each family, i.e., some articles scored > 1.

**Figure 4 biology-13-00610-f004:**
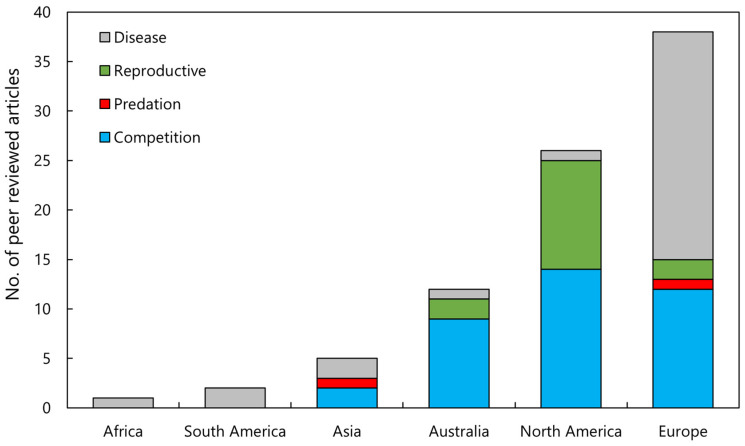
Analysis of the literature investigating the impact of invasive crayfish on native crayfish across six continents and how many of the articles investigated the four mechanisms of impact (*n* = 83).

**Figure 5 biology-13-00610-f005:**
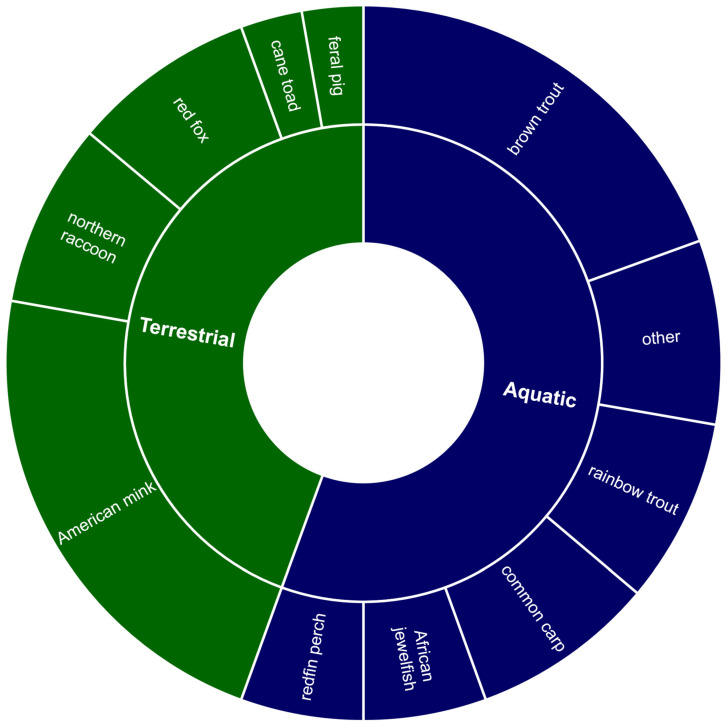
Summary of the literature investigating the impacts of non-native species (other than invasive crayfish) on native freshwater crayfish (*n* = 33). The inner ring quantifies the proportion of terrestrial or aquatic/amphibious species appearing in the literature. The outer ring indicates the proportion of the literature focused on each non-native species.

**Figure 6 biology-13-00610-f006:**
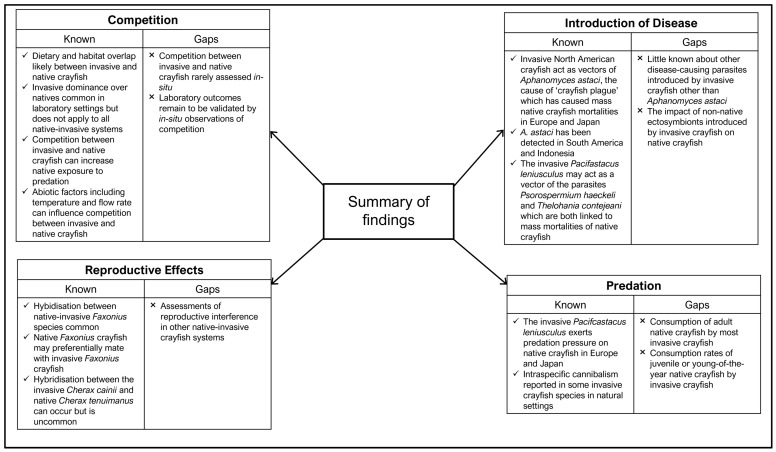
Summary of the current knowledge surrounding the impacts of invasive crayfish on native crayfish as well as the knowledge gaps identified in the literature.

**Figure 7 biology-13-00610-f007:**
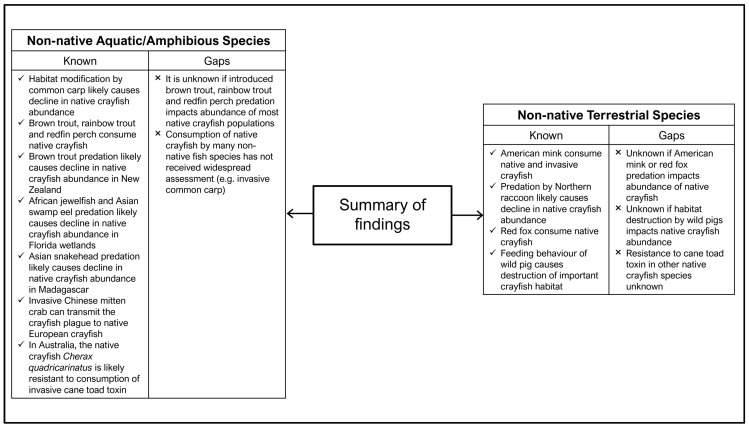
Summary of the current knowledge surrounding the impacts of non-native aquatic/amphibious and terrestrial species on native crayfish as well as the knowledge gaps identified in the literature.

## Data Availability

Not applicable.

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
