# Peer review of "The Impacts of Invasive Crayfish and Other Non-Native Species on Native Freshwater Crayfish: A Review"

_biology, 2024, doi:10.3390/biology13080610_

Round 1
Reviewer 1 Report
Comments and Suggestions for Authors
Dear authors, the Review “The Impacts of Invasive Crayfish ...” is of undoubted interest. The authors have done a lot of work by presenting an overview of the articles available in the Web of Science and Google Scholar databases and attempted to analyze the available data. The shortcomings of this manuscript include those in the discussion of the material, the design of the text and figures, and most importantly – in the material included.
1. First of all, what is surprising is the relatively short list of literature that the authors used for the review, which is a total of 172 publications. I assume that the search approach has some drawbacks, or the authors are not thorough enough. For example, searching for the keywords suggested by the authors, “alien crayfish”, in a conventional search engine immediately found several publications that were not included in this review. These studies overlooked by the authors included even large monographs, for example, "Invasive Aquatic Species of Europe. Distribution, Impacts and Management", which has a dedicated chapter on "Alien Crayfish in Europe: Negative and Positive Impacts and Interactions with Native Crayfish". Also there are many articles in scientific journals, here are some of them.
· Yiyong Chen, Changsen Sun, Aibin Zhan “Biological Invasions in Aquatic Ecosystems in China” (Aquatic Ecosystem Health and Management 2017. 10(4):402-412. DOI: 10.1080/14634988.2017.1403268)
· Nunes, Ana & Zengeya, Tsungai & Hoffman, Andre & Measey, John & Weyl, Olaf. (2017). Distribution and establishment of the alien Australian redclaw crayfish, Cherax quadricarinatus, in South Africa and Swaziland. PeerJ. 5. e3135. 10.7717/peerj.3135.
· Kawai, T., Scholtz, G., Morioka, S., Ramanamandimby, F., Lukhaup, C., & Hanamura, Y. Parthenogenetic Alien Crayfish (Decapoda: cambaridae) Spreading in Madagascar (2009). Journal of Crustacean Biology, 29(4), 562-567.
· Martin Weinländer and Leopold Füreder Native and alien crayfish species: do their trophic roles differ? (Freshwater Science. 2016. Volume 35, No 4). DOI https://doi.org/10.1086/689031)
This is a tiny fraction of the articles that were not included in this review, I found them in passing, in "two clicks", and so I assume that a serious search will yield a large array of data. The authors may object that they had a single principle of searching – using two systems (WoS and GS). In this case, it is necessary to change the title so that the reader can adequately assess the reliability of the information presented in the review.
2. In the discussion of the material, the study of the impact of invasive crayfish on native crayfish across six continents is compared, but they forgot to elaborate that the number and distribution of freshwater crayfish in South America are limited, and the fauna of freshwater crayfish in Africa is restricted to Madagascar, because there are no crayfish in the rivers of the African continent proper (only freshwater crabs that this review does not cover). What harm is there for the native species if there are no such species to begin with?
3. A similar note on the extensiveness of study of different groups of crustaceans, Astacidae, Cambaridae, Parastacidae and different species. How is the extensiveness of study of different groups is related to their abundance, to the amount and distribution of invasive species? I would like to see at least some suggestions. Is it safe to assume that widespread species have been studied more?
4. The review mentions alien species of terrestrial and aquatic animals from other taxa that also can harm native crayfish populations; and while this is not directly related to the topic, this is a very interesting aspect. The authors could compare the negative impact of different invaders, but they did not. Unfortunately, this part of the review is even less well done than the main one, and needs significant additions.
5. Design of the figures. It is not common practice to mention the type of diagram in the name of the figure, as it was done in Fig. 2, 5 (Donut chart…). The reader of a scientific article is usually able to determine the type of diagram by himself.
6. Fig. 5 has some flaws in its design. Species names are sometimes capitalized and sometimes lowercase; also in Fig. 2 the Latin names of the species were used, while in Fig. 5 the English ones. Why?
7. The quotation in the text does not follow the rules of the journal – round brackets and the names of the authors, instead of square brackets and numbers.
I suggest that the authors should supplement the review with additional sources on this topic, and address the shortcomings in the design. The manuscript needs a more complete review of the literature and a more accurate analysis and interpretation of the results.
Kind regards
Author Response
Thank you for taking the time to review this manuscript. Please find the detailed responses below and the corresponding revisions/corrections highlighted/in track changes in the re-submitted files.
Comment 1: First of all, what is surprising is the relatively short list of literature that the authors used for the review, which is a total of 172 publications. I assume that the search approach has some drawbacks, or the authors are not thorough enough. For example, searching for the keywords suggested by the authors, “alien crayfish”, in a conventional search engine immediately found several publications that were not included in this review. These studies overlooked by the authors included even large monographs, for example, "Invasive Aquatic Species of Europe. Distribution, Impacts and Management", which has a dedicated chapter on "Alien Crayfish in Europe: Negative and Positive Impacts and Interactions with Native Crayfish". Also there are many articles in scientific journals, here are some of them.
- Yiyong Chen, Changsen Sun, Aibin Zhan “Biological Invasions in Aquatic Ecosystems in China” (Aquatic Ecosystem Health and Management 2017. 10(4):402-412. DOI: 10.1080/14634988.2017.1403268)
- Nunes, Ana & Zengeya, Tsungai & Hoffman, Andre & Measey, John & Weyl, Olaf. (2017). Distribution and establishment of the alien Australian redclaw crayfish, Cherax quadricarinatus, in South Africa and Swaziland. PeerJ. 5. e3135. 10.7717/peerj.3135.
- Kawai, T., Scholtz, G., Morioka, S., Ramanamandimby, F., Lukhaup, C., & Hanamura, Y. Parthenogenetic Alien Crayfish (Decapoda: cambaridae) Spreading in Madagascar (2009). Journal of Crustacean Biology, 29(4), 562-567.
- Martin Weinländer and Leopold Füreder Native and alien crayfish species: do their trophic roles differ? (Freshwater Science. 2016. Volume 35, No 4). DOI https://doi.org/10.1086/689031)
This is a tiny fraction of the articles that were not included in this review, I found them in passing, in "two clicks", and so I assume that a serious search will yield a large array of data. The authors may object that they had a single principle of searching – using two systems (WoS and GS). In this case, it is necessary to change the title so that the reader can adequately assess the reliability of the information presented in the review.
Response 1: Thank you for the thorough assessment of the literature assessed in the manuscript. However, it is clear there has been a significant misunderstanding of the aims and scope of the review. We feel it’s possible that reviewer 1 may have interpreted the aim of the review to be an assessment of the impacts of invasive crayfish. We emphasise here that the aim of this review is to more specifically examine the mechanisms by which invasive crayfish impact native crayfish species and examine the mechanisms by which other introduced species impact native crayfish species. These aims are clearly stated in lines 125-127. Therefore, we have only included literature that examines or documents one or more impact(s) of an invasive crayfish or other non-native species on a native crayfish. This is stated in lines 150-151.
We feel there has been a misunderstanding of these aims for two reasons. (1) The keywords that reviewer 1 used in your literature search “alien crayfish” were not keywords used in isolation in our literature search methods. It is emphasized in our methods section that the search term “alien crayfish” was searched alongside the terms “impact” or “threat” or “compete” or “disease” or “predate” or “consume” or “reproductive interference” or “hybridization” AND “native freshwater crayfish” or “indigenous freshwater crayfish”. Please see lines 151-155. The inclusion of these additional terms is critical in yielding literature relevant to aims and scope of the review. Simply searching the term “alien crayfish” yields hundreds of articles that are beyond the scope of this review. (2) While the book chapter “Alien Crayfish in Europe: Negative and Positive Impacts and Interactions with Native Crayfish” is certainly relevant and we have now included this in our list of previous reviews to examine the impact of invasive crayfish on native crayfish Europe (see line 122), the other four articles listed are out of scope of the review. Below is a response to each paper regarding the reasons each is out of scope:
Yiyong Chen, Changsen Sun, Aibin Zhan “Biological Invasions in Aquatic Ecosystems in China” (Aquatic Ecosystem Health and Management 2017. 10(4):402-412. DOI: 10.1080/14634988.2017.1403268)
This article reviews the invasive aquatic species in China, summarises the broad negative impacts of invasive aquatic species in China and the major vectors of these species. This article does not document a specific impact of an invasive crayfish or other non-native species on a native freshwater crayfish. Notably, China does not have any native species of freshwater crayfish, therefore, this paper cannot be relevant to the scope of the review.
Nunes, Ana & Zengeya, Tsungai & Hoffman, Andre & Measey, John & Weyl, Olaf. (2017). Distribution and establishment of the alien Australian redclaw crayfish, Cherax quadricarinatus, in South Africa and Swaziland. PeerJ. 5. e3135. 10.7717/peerj.3135.
This article documents the distribution, rate of spread and population dynamics of Cherax quadricarinatus in South Africa and Swaziland. This article does not document an impact of this invasive crayfish on a native crayfish, further, no native crayfish occur in continental Africa, therefore, this article cannot be relevant to the scope of the review.
Kawai, T., Scholtz, G., Morioka, S., Ramanamandimby, F., Lukhaup, C., & Hanamura, Y. Parthenogenetic Alien Crayfish (Decapoda: cambaridae) Spreading in Madagascar (2009). Journal of Crustacean Biology, 29(4), 562-567.
This article documents the taxonomy of the invasive marbled crayfish in Madagascar. While it very briefly mentions this species may impact native crayfish in the “conservation perspective” section, this article does not document a specific impact of this invasive crayfish on a native crayfish and is therefore out of scope of this review.
Martin Weinländer and Leopold Füreder Native and alien crayfish species: do their trophic roles differ? (Freshwater Science. 2016. Volume 35, No 4). DOI https://doi.org/10.1086/689031)
This article examines trophic differences between native Astacus astacus and Austropotamobius torrentium and the invasive Pacifastacus leniusculus. While differences in the trophic roles of the two species are found, trophic interactions between the native and invasive species are not examined as part of this study, and therefore this article does not document or examine an impact of the invasive crayfish on the native crayfish species. Therefore, this article is out of scope of the review.
We therefore wish to assure that our search of the relevant literature was extremely thorough. Not only did we search for literature using Web of Science and Google Scholar, but we also examined each reference list of each relevant article found for additional articles relevant to the review (this is stated in lines 157-158). If we were to expand the scope of the review to such an extent that would include the above articles, the manuscript may be double its current length. We also wish to emphasise that the title of our manuscript “The Impacts of Invasive Crayfish and Other Non-Native Species on Native Freshwater Crayfish: A Review” is clear in communicating the contents and with reflection we do not believe the title can be any more explicit than it currently is. We also note that 172 publications is not a small amount of literature for inclusion in a review paper.
Comment 2: In the discussion of the material, the study of the impact of invasive crayfish on native crayfish across six continents is compared, but they forgot to elaborate that the number and distribution of freshwater crayfish in South America are limited, and the fauna of freshwater crayfish in Africa is restricted to Madagascar, because there are no crayfish in the rivers of the African continent proper (only freshwater crabs that this review does not cover). What harm is there for the native species if there are no such species to begin with?
Response 2: Agreed, thank you for drawing our attention to this. The amount of literature is certainly influenced by the distribution of freshwater crayfish across the continents. We have added the following statement to lines 193-199 [“This distribution of literature across the continents is likely related to both the number and abundance of native crayfish species, the presence of invasive crayfish on each continent and the number of mechanisms through which the invasive crayfish impact natives. For example, in Africa only seven native crayfish species occur all of which are isolated to Madagascar and only one invasive crayfish, Procambarus virginalis, has been introduced to Madagascar [63]. Therefore, the literature examining the impacts of invasive crayfish on native crayfish in Africa is limited (Figure 4).”]
Comment 3: A similar note on the extensiveness of study of different groups of crustaceans, Astacidae, Cambaridae, Parastacidae and different species. How is the extensiveness of study of different groups is related to their abundance, to the amount and distribution of invasive species? I would like to see at least some suggestions. Is it safe to assume that widespread species have been studied more?
Response 3: Agreed, this is also a factor that is likely to substantially influence the amount of research and literature on each species. We have added a statement in lines 186-187 [“This is a likely consequence of how widespread these invasive species are relative to other invasive crayfish”].
Comment 4: The review mentions alien species of terrestrial and aquatic animals from other taxa that also can harm native crayfish populations; and while this is not directly related to the topic, this is a very interesting aspect. The authors could compare the negative impact of different invaders, but they did not. Unfortunately, this part of the review is even less well done than the main one, and needs significant additions.
Response 4: It’s unclear why it has been interpreted that the impact of alien terrestrial and aquatic animals is not related to the topic. It is stated in the title of the review “The Impacts of Invasive Crayfish and Other Non-Native Species on Native Freshwater Crayfish: A Review” and is clearly stated in the manuscript that as well as examining the impact of invasive crayfish on native crayfish we aim to examine the mechanisms by which other introduced species impact native crayfish species (see lines 126-127). Therefore, this section of the review is an integral component and as mentioned in lines 110-111, there has yet to be a review of the impacts other non-native species on native crayfish. Regarding the comparison of different invaders, we note that we do provide an assessment of which mechanisms of impact are most documented in the literature for invasive crayfish and other non-native species in the “Trends in the Literature” section (see lines 187-190 and lines 205-208). However, any further comparative assessments does not serve to address the aims of the review and therefore was not and will not be included in this paper. As other “significant additions” that are required to improve this section have not been specified, we cannot comment on other elements that may be perceived to be missing from this section. We wish to assure, that the literature on the impacts of other non-native species was searched as thoroughly as is it was for the “Impacts of invasive crayfish on Native Crayfish” section.
Comment 5: Design of the figures. It is not common practice to mention the type of diagram in the name of the figure, as it was done in Fig. 2, 5 (Donut chart…). The reader of a scientific article is usually able to determine the type of diagram by himself.
Response 5: Agreed. The figure captions have now been updated to [“Summary of the literature…”]. See line 226 and 240.
Comment 6: Fig. 5 has some flaws in its design. Species names are sometimes capitalized and sometimes lowercase; also in Fig. 2 the Latin names of the species were used, while in Fig. 5 the English ones. Why?
Response 6: Thank you for drawing our attention to this oversight, this has now been rectified (see line 239). We elected to use the common names of these species since they are better known by their common names, while crayfish are often better known by their scientific names.
Comment 7: The quotation in the text does not follow the rules of the journal – round brackets and the names of the authors, instead of square brackets and numbers.
Response 7: The citation style has now been updated.
Reviewer 2 Report
Comments and Suggestions for Authors
The manuscript is well-done, and I recommend to publish one after a minor revision. But see the notes to the authors, this is my plan for the MS inproving:
Title, and below
1) At first record of a taxonomic group, the authors need to represent an exact taxonomic position (Decapoda: Cambaridae, Astacidae and Parastacidae) in the title, Abstract and main text.
Introduction
2) I guess that the authors need to record pioneer works on biological invasions, inckuding thise in continental water bodies, in Introduction.
3) The same for invasive crayfish… “First observations of the crayfish invasions are dated back to…”.
Material and Methods
4) During definition of introduced species, invasive species, non-native species, the authors need to represent references to previous authors with similar ideas.
5) The authors need to discuss, are their number of 83 papers enough for their conclusions?
Results
6) “Although most of the literature reports invasive crayfsh as competitively dominant over natives, there are some exceptions to this fnding”. Are You sure that this is not an artifact of a “negative results” bias (when only positive results are published, but negative results are regarded as useless). It means that neutral reports were not published…
Discussion…
7) To date, the Discussion section is concentrated on the crayfish only. We do not know, are crayfishes unique, or no? I think that the authors need to add at least a large paragraph discussion similarities and dissimilarities with other invasive crustaceans (and other invertebrates?) in continental waters.
Great success to the authors!
Author Response
Thank you for taking the time to review this manuscript. Please find the detailed responses below and the corresponding revisions/corrections highlighted/in track changes in the re-submitted files.
Comment 1: At first record of a taxonomic group, the authors need to represent an exact taxonomic position (Decapoda: Cambaridae, Astacidae and Parastacidae) in the title, Abstract and main text.
Response 1: Thank you for this suggestion, however, we do not feel this is a necessary addition to the title and abstract given that this review is not targeting one specific taxonomic group of freshwater crayfish, but reviewing the literature on all freshwater crayfish. Additionally, since the review not only focuses on the impacts of invasive crayfish but other non-native species too, if listing the taxonomic positions of crayfish, it would be necessary to list to taxonomic positions of the other non-native species We view this as impractical and clumsy to include such a list in the title. We have listed the crayfish families in the Results and Discussion section, please see lines 183-184.
Comment 2: I guess that the authors need to record pioneer works on biological invasions, inckuding thise in continental water bodies, in Introduction.
Response 2: Agreed. Some pioneer works on biological invasions already cited in the manuscript (e.g. Sakai et al., 2001; Kolar & Lodge, 2000), however, we have now include a citation of what is considered the seminal work in biological invasions “The Ecology of Invasions by Animals and Plants” by Charles Elton. Please see line 68.
Comment 3: The same for invasive crayfish… “First observations of the crayfish invasions are dated back to…”.
Response 3: Agreed. We have now added [“The first accounts of crayfish introductions beyond their natural range date back to 1746 [30]”] to lines 89 and 90.
Comment 4: During definition of introduced species, invasive species, non-native species, the authors need to represent references to previous authors with similar ideas.
Response 4: Agreed. We have now cited the work on which this definition was based. Please see line 138.
Comment 5: The authors need to discuss, are their number of 83 papers enough for their conclusions?
Response 5: In this review it is not possible to draw final conclusions about the impact invasive crayfish and other non-native species on native crayfish but rather review and document the impacts that are present in the literature. Further, the distribution of these articles across the four mechanisms of impact (competition, predation, reproductive effects and introduction of disease) is uneven as documented in Figure 3. This allowed us to identify and make explicit comment on areas where literature is lacking in the “Knowledge Gaps and Future Research” section. No change made.
Comment 6: “Although most of the literature reports invasive crayfsh as competitively dominant over natives, there are some exceptions to this fnding”. Are You sure that this is not an artifact of a “negative results” bias (when only positive results are published, but negative results are regarded as useless). It means that neutral reports were not published…
Response 6: The reviewer has identified a universal issue in scientific literature. In short, it is a very good point and argues that we must be cautious in our interpretation of the literature. We believe that our statements in the text are sufficiently cautious and contend that no further change to the text are necessary: “most of the literature reports invasive crayfish as competitively dominant over natives, there are some exceptions to this finding”. No change made.
Comment 7: To date, the Discussion section is concentrated on the crayfish only. We do not know, are crayfishes unique, or no? I think that the authors need to add at least a large paragraph discussion similarities and dissimilarities with other invasive crustaceans (and other invertebrates?) in continental waters.
Response 7: It is unclear what is meant here. If you are referring to the impacts of non-native species other than invasive crayfish, there is an entire section of the review dedicated to discussing the impacts of these species. Please see lines 462-599. In our assessment of the literature only one article was found documenting an impact of an invasive invertebrate (other than invasive crayfish), the Chinese mitten crab (see lines 543-549). We feel the existence of just one article does not warrant an entire paragraph comparing the impacts of invasive crayfish and other invasive invertebrates on native crayfish.
Round 2
Reviewer 1 Report
Comments and Suggestions for Authors I thank the authors for their detailed comments. After that, I began to understand their logic better. Minor flaws, Comments 5-7, have been fixed, with the exception of fig.5 design, Species names are sometimes capitalized and sometimes lowercase, but it doesn't really matter. I agree that the manuscript can be published.Kind regards
Author Response
Thank you again for your valuable comments that have contributed to the improvement of our manuscript. Please see below a reponse to your comment.
Comment 1: Minor flaws, Comments 5-7, have been fixed, with the exception of fig.5 design, Species names are sometimes capitalized and sometimes lowercase, but it doesn't really matter. I agree that the manuscript can be published.
Response 1: The two species names that have been capitalised in figure 5 are 'American mink' and 'African jewelfish'. The reason these are capitalised and others aren't is because it is correct grammar to capitalise a proper noun (i.e. a specific place) if present in a species name. No change.